# A Transcriptomic Response to *Lactiplantibacillus plantarum*-KCC48 against High-Fat Diet-Induced Fatty Liver Diseases in Mice

**DOI:** 10.3390/ijms23126750

**Published:** 2022-06-17

**Authors:** Ilavenil Soundharrajan, Muthusamy Karnan, Jeong-Sung Jung, Kyung-Dong Lee, Jeong-Chae Lee, Thiyagarajan Ramesh, Dahye Kim, Ki-Choon Choi

**Affiliations:** 1Grassland and Forage Division, Rural Development Administration, National Institute of Animal Science, Cheonan 31000, Korea; ilavenil@korea.kr (I.S.); karnan7899@rediffmail.com (M.K.); jjs3873@korea.kr (J.-S.J.); 2Department of Companion Animals, Dongsin University, Naju 58245, Korea; leekd@dsu.ac.kr; 3Department of Bioactive Material Sciences and Research Center of Bioactive Materials, Jeonbuk National University, Jeonju 54896, Korea; leejc88@jbnu.ac.kr; 4Department of Basic Medical Sciences, College of Medicine, Prince Sattam Bin Abdulaziz University, Al-Kharj 11942, Saudi Arabia; thiyagaramesh@gmail.com; 5Animal Genomics and Bioinformatics Division, National Institute of Animal Science, Wanju 55365, Korea

**Keywords:** high-fat diet, fatty liver diseases, *L. plantarum*, transcriptome

## Abstract

The most prevalent chronic liver disorder in the world is fatty liver disease caused by a high-fat diet. We examined the effects of *Lactiplantibacillus plantarum*-KCC48 on high-fat diet-induced (HFD) fatty liver disease in mice. We used the transcriptome tool to perform a systematic evaluation of hepatic mRNA transcripts changes in high-fat diet (HFD)-fed animals and high-fat diet with *L. plantarum* (HFLPD)-fed animals. HFD causes fatty liver diseases in animals, as evidenced by an increase in TG content in liver tissues compared to control animals. Based on transcriptome data, 145 differentially expressed genes (DEGs) were identified in the liver of HFD-fed mice compared to control mice. Moreover, 61 genes were differentially expressed in the liver of mice fed the HFLPD compared to mice fed the HFD. Additionally, 43 common DEGs were identified between HFD and HFLPD. These genes were enriched in metabolic processes, retinol metabolism, the PPAR signaling pathway, fatty acid degradation, arachidonic metabolism, and steroid hormone synthesis. Taking these data into consideration, it can be concluded that *L. plantarum-*KCC48 treatment significantly regulates the expression of genes involved in hepatosteatosis caused by HFD, which may prevent fatty liver disease.

## 1. Introduction

Obesity is one of the major health issues we face today. Human diseases such as cardiovascular diseases, insulin resistance, diabetes mellitus, and nonalcoholic fatty liver disease (NAFLD) are closely linked to fat deposition and metabolism [1]. If post-2000 trends continue, global obesity will reach 18% for men and 21% for women by 2025. The prevalence of severe obesity will reach 6% in men and 9% in women [2]. The liver has numerous metabolic functions, including glucose and lipid metabolism, bile salt synthesis, detoxification of xenobiotic compounds, and secretion of plasma proteins [3]. It is highly regulated by nutritional and hormonal factors in the body to maintain nutrient and energy homeostasis. Non-alcohol fatty liver diseases (NAFLD), or hepatic steatosis, is closely associated with obesity [4]. High-fat diets result in extensive changes in the liver, leading to nonalcoholic fatty liver disease, insulin resistance, dyslipidemia, and obesity [5]. Metabolic disorders resulting in fat accumulation are caused primarily by insulin resistance. Furthermore, hormone-sensitive lipase activity in adipocytes is suppressed because of insulin resistance, while triglycerides released by adipocytes are converted into free fatty acids and subsequently increase fatty acids entering the liver, resulting in fatty liver.

A critical strategy is needed to protect the liver from the damage caused by fat metabolism, as the prevalence of fatty liver diseases has increased. Several researchers have been focused on the development of dietary supplements to balance the excess energy input caused by the overconsumption of rice foods [6]. One of the most actively studied sources of anti-obesity efficiency is probiotics [7,8]. Probiotics have been proposed as an anti-obesity agent through several molecular mechanisms, including metabolic changes [9,10], improvement of the intestinal barrier, modulating the immune response [11], reduced adipocyte size [10], and decreasing dietary fat absorption [12]. Supplementation of probiotics to high-fat diet-induced obese mice alleviates body weight gain and adiposity by modulating the composition of the gut-associated microbiota. Probiotics and/or prebiotics are effective in lowering serum/lipids levels [10]. In animal models, *lactobacillus* species exhibited potential probiotic and hypocholesterolemia effects [13]. A number of studies have examined the effects of probiotics on diet-induced NAFLD in animal models. It has been proven that probiotic supplementation, specifically *Lactobacillus* and *Bifidobacterium,* can prevent diet-induced fatty liver diseases through downregulation of lipogenesis, reactive oxygen species, proinflammatory markers and mediators, as well as lipopolysaccharide (LPS) and Toll-like receptor-4 (TLR-4). LPS triggers cytokine cascades and inflammation by interacting with TLR-4. In addition, probiotics increase fatty acid oxidation, antioxidant activity, insulin sensitivity, and intestinal mucosal integrity, as well as modulate gut microbiota and bile acid metabolism [14,15]. The effects of *lactobacillus* species on body weight change vary depending on the host. *L. plantarum* (new taxonomy name *L. plantarum* [16]) has gained a lot of attention among *lactobacilli* for its biological potential as a probiotic. It is considered the safest probiotic (GRAS) with a qualified Presumption of Safety (QPS) status and has a long history. *L. plantarum* inhibits inflammation, dyslipidemia, hypocholesterolemia, insulin resistance, and obesity, as well as modulates gut microbiota [9,17]. *L. plantarum* reduced the fat percentage in healthy volunteers as well as the size of adipocytes in mice. Furthermore, it reduced the size of adipocytes, which in turn reduced the effects of diet-induced obesity [18]. Recently, we studied the anti-obesity activity of *L. plantarum*-KCC48 in high-fat diet-induced obese mice and its probiotic potentials, which suggested that the *L. plantarum*-KCC48 inhibited adipocyte differentiation and lipid accumulation in 3T3-L1 adipocytes. *L. plantarum* supplementation reduced fat mass and serum lipid profile concurrently with the downregulation of lipogenic gene expression in adipocytes, resulting in a reduction in bodyweight of HFD-fed obese mice through activation of p38MAPK, p44/42, and AMPK-α by increasing their phosphorylation and modulating gut-associated microbiota [10]. Evidence suggested that changing gut-associated microbiota via a diet rich in probiotics can be an effective approach to the treatment of obesity-induced metabolic diseases and disorders. In our experience, probiotics alleviate diet-induced obesity by regulating different signaling pathways. Transcriptome sequencing methods are integral to research on these signaling pathways. Differential gene expression (DGE) is a high-throughput transcriptome method that has become an integral part of many genomic studies of diseases and biological processes [19]. It has higher throughput, sensitivity, and economics compared to conventional transcriptome analysis [20]. Next-generation sequencing technologies were used to examine the effect of *Lactiplantibacillus plantarum*-KCC48 on fatty liver disease in mice fed a high-fat diet. Differential gene expressions were identified in experimental tissues and we studied the biological roles of differentially expressed genes.

## 2. Results

We examined the impact of *Lactiplantibacillus plantarum-KCC48* as a probiotic on hepatic transcriptomic changes in high-fat diet-induced fatty liver disease in mice using Next-Generation Sequencing (NGS). Experiments were conducted on mice fed a standard diet, HFD diet, and HFLPD diet for eight weeks. Animals fed different diets displayed no abnormal behaviors during the experiment. Mice fed the HFD diet had higher body weight than control mice, and mice fed the HFLPD had lower body weight than HFD-fed mice. Furthermore, liver markers such as aspartate transaminase (AST), alanine transaminase (ALT), and lipid profiles such as total cholesterol, triglycerides, and LDL were higher in HFD-fed sera samples than in control animals, while animals treated with HFLPD had significantly lowered liver markers and lipid profiles almost to normal levels [10]. It is interesting to note that this finding was strongly supported by the results of the weights of liver in experimental animals, which showed that HFD-fed mice had higher liver weight and TG content compared with control mice, whereas liver weight and TG content declined significantly in mice supplemented with HFLPD (Figure 1a,b).

### 2.1. Transcriptome Validation ASSESSMENT

The global gene expression changes in the high-fat diet-fed (HFD) animals and the high-fat diet with probiotic *L. plantarum*-KCC48-fed animals (HFLPD) were determined by the RNA sequence tool with three replicates per group. In accordance with our expectations, HFD-fed animals had significantly higher triglyceride levels and liver weight. Samples of RNA sequence data showing gene expression reads expressed as fragments per kilobase of transcript per million mapped reads (FPKM) were highly reliable and reproducible. PCA of all experimental groups showed almost identical samples within each group (Figure 1c). Overall, the RNA sequencing results were consistent and reliable across all experimental samples.

### 2.2. Overview of Transcriptome Changes

In order to identify differentially expressed genes (DEGs) in experimental groups, we used fold changes greater than two and *p*-values less than 0.05. Figure 1d shows the total number of differentially expressed genes in the liver for the HFD- and HFLPD-fed animals. Animals with HFD had 145 differentially expressed genes in liver tissue compared to animals fed with a normal diet. Of the total number of genes expressed, 105 genes (72.41%) were upregulated while 40 genes (27.59%) were downregulated (*p* < 0.05) in HFD-fed animals compared with animals from the control group (Figure 1d,e and see Appendix A). However, animals fed with HFLPD were found to have 62 differentially expressed genes (*p* < 0.05) in the liver tissue compared to animals fed with HFD. There were 36 genes (58.06%) upregulated and 26 genes (41.94%) downregulated (Figure 1d,f and see the Appendix A). The differentially expressed genes in HFD liver were mainly enriched in the GO terms extracellular matrix (1.17%), aging (0.34%), angiogenesis (1.29%), and neurogenesis (0.47%), followed by immune and inflammatory responses, cellular migration, cell differentiation, cell death, and apoptotic processes (Figure 2a). In HFLPD liver tissues, DEGs were more actively enriched with RNA splicing (0.31%), immune response (0.32%), apoptosis (0.36%), cell differentiation (0.26%), cell death (0.33%), cell migration (0.34%), and inflammatory response (0.21%), among others (Figure 2b).

### 2.3. Location of Differentially Expressed Genes in Liver

The majority of differentially expressed genes are found in the endoplasmic reticulum (22%) and its membrane (19%), extracellular exosomes (19%), intracellular membrane-bounded organelles (17%), organelle membranes (14.7%), the extracellular region (14%), and extracellular space (10%), and fewer genes were located in the basolateral plasma membrane, the MHC class II protein complex, the integral component of the endoplasmic reticulum membrane, endoplasmic reticulum lumen, and high-density lipoprotein particles (Figure 2c).

### 2.4. Common Differentially Expressed Genes between HFD and HFLPD Groups

We then analyzed the common DEGs between HFD and HFLPD. Both HFD and HFLPD shared 43 DEGs (Table 1). Both genes were detected in both groups involved in the contraregulation of biological processes. A total of 43 transcripts were identified in liver tissues between HFD and HFLPD that were associated with the contraregulation of biological process. Of these, 34 genes were significantly upregulated and 9 genes were downregulated in HFD-fed mice, while 34 genes were downregulated and 9 were upregulated in response to HFLPD treatment. DEGs detected between HFD and HFLPD have been shown to play major roles in the GO terms retinol metabolism, PPAR signaling pathway, fatty acid degradation, arachidonic metabolism, and steroid hormone synthesis (Figure 3a,b). In addition, we calculated and plotted contraregulated genes in both groups of liver tissues (Figure 3c).

###  2.5.  Functional Characterization of DEGs in HFD and HFLPD

By using DAVID tool analysis, DEG functional annotations were identified. More than ten counts were used to detect functional characterizations. DEGs in HFD were associated with more than 40 biological functions, including steroid hormone biosynthesis; cholesterol, lipid, and cholesterol metabolism; endoplasmic reticulum; cytochrome p450; and disulfide bond (Table 2). The DEGs identified in HFLPD liver tissues are closely associated with 28 biological functions, including membrane, metal binding, endoplasmic reticulum, disulfide bond, glycoprotein, oxidoreductase activity, metabolic pathways, retinol metabolism, cytochrome p450, and monooxygenase (Table 3).

###  2.6.  KEGG Signaling Enrichment Analysis for DEGs in HFD and HFLPD

Next, we identified the KEGG signaling pathways of DEGs in each group. DEGs identified in HFD-fed liver tissue were associated with 27 KEGG signaling pathways compared to control group animals. The DEGs were closely associated with metabolic pathways (27.6%), retinol metabolism (12.4%), PPAR signaling (9%), chemical carcinogenesis (8.3%), antibiotic biosynthesis (7.6%), steroid hormone biosynthesis (6.9%), etc. (Table 4). HFLPD DEGs are mainly enriched for metabolic pathways (28.3%), retinol metabolism (20%), the PPAR signaling pathway (13.3%), chemical carcinogenesis (13.3%), arachidonic acid metabolism (11.7%), steroid hormone biosynthesis (10%), etc. (Table 5).

##  3.  Discussion

A major role of the liver includes the synthesis, storage, and redistribution of lipids, amino acids, and glucose under highly coordinated and dynamic conditions regulated by dietary intake, environment circadian rhythms, and hormonal and neuronal stimulations [21,22]. Physiological dysfunction of the liver can lead to insulin resistance and type II diabetes [23]. Non-alcoholic fatty liver disease results from an excess deposition of fat in hepatocytes that progresses from simple liver steatosis to non-alcoholic steatohepatitis (NASH) and, in more severe cases, liver fibrosis, cirrhosis, and hepatocarcinoma [24]. Probiotics have been used in clinical and medical fields to treat intestinal diseases, renal complications, lung, brain, and cardiovascular diseases. *Lactiplantibacillus plantarum* shows strong hepatoprotective activity against alcoholic liver disease [20,25,26] and NAFLD [27,28,29]. We performed global gene regulations in fatty liver depositions in mice fed either a high dietary fat diet or a high dietary fat diet containing *L. plantarum* through comprehensive analysis of transcriptome data, and we identified differentially expressed genes (DEGs) in all groups and mapped them to GO and KEGG databases and then further compared their expression to pathways. We identified 43 DEGs that were common in both HFD and HFLPD and showed contraregulation in the liver of both groups. In addition, we compared the results from functional annotations and KEGG pathways for more specific changes in HFD- and HFLPD-fed liver tissues. Using functional annotation clustering analysis, major DEGs identified in HFD-fed animal liver were closely related to membrane, signal, disulfide bond, metabolic pathways, glycoprotein, signal peptide, endoplasmic reticulum, acetylation, extracellular exosome, and metal binding activity. Furthermore, significant numbers of DEGs were also enriched with lipid metabolism, lipid biosynthesis, cholesterol and fatty acid metabolism, and steroid synthesis. DEGs found in HFLPD liver tissue had several biological functions, but most of them were closely related to processes other than fat metabolism.

HFD-fed animals can induce changes in liver biological processes, which can be significantly attenuated with *L. planatarum* supplementation. The most common method for inducing NALFDs is the Western diet [30] which has high saturated fat, trans-fat, and sugar content. Diets of this type could result in obesity, metabolic syndrome, NAFLD, and NASH in humans [31]. HFD-fed mice showed increased levels of free fatty acids, insulin resistance, reduced fatty acid oxidation, and increased de novo lipogenesis [32]. This was highly consistent with the present study, as we found an increase in total liver weight and triglyceride content in the liver tissue of HFD-fed animals compared to control animals, whereas these abnormal changes were significantly reversed in animals fed with HFLPD. In the study, it was shown that the probiotic could contribute to the improvement of animal health of HFD-fed animals through the regulation of several molecular and metabolic pathways.

Multiple cytochrome P450 (CYPs) family genes have been associated with high-fat diet-induced liver disease. CYPs are closely related to the liver’s metabolism of drugs, chemicals, and other endogenous substrates. Additionally, CYPs have been implicated in the pathogenesis of several liver diseases. We identified several DEGS for CYPs in liver tissues in the present study, including CYP2a, CYP2b, CYP2c, CYP3a, CYP4a, and CYP7. Each has several isoforms and unique activities in the liver. CYP2A12 is the bile acid 7a hydroxylase that converts secondary deoxycholic acid (DCA) and lithocholic acid (LCA) into primary cholic acid (CA) and chenodeoxycholic acid (CDCA), respectively [33]. Mice supplemented with a high-fat diet showed an increase in CYP2a12 and CYP2a22 activity. On the other hand, in the present study, the DEG of CYP2a12 in mice livers with HFD-induced obesity was downregulated, while that of CYP2a22 was upregulated by HFLPD supplements. A high-fat diet upregulated the expression of CYP2b9, the most highly inducible gene closely associated with obesity [34,35]. HFD-fed animals showed significant increases in DEG of CYP2b9 expression, while HFLPD animals showed reversed expression. This finding was in line with previous research. In male mice, CYP2c deficiency decreased muricholic acids that protect against obesity caused by high-fat diets, while at the same time promoting liver damage [36]. Xiang et al. reported that Cyp2c38 and Cyp2c40 were increased in db/db mice and decreased in DEX-treated mice [37]. In mice treated with an atherogenic diet, Cyp2c39 was upregulated [38]. Our results showed that DEGs of Cyp2c38 and Cyp2c39 were highly expressed in HFD-fed animal liver, whereas the expression of Cyp2c38 and Cyp2c39 genes was downregulated in livers fed with HFLPD. CYP7A11 is a mouse homolog of CYP3A4 involved in the metabolism of the hypnotic drug midazolam. After high-fat feeding of mice, CYP7A11 expression was decreased in liver [39,40]. On the other hand, Cyp3a11 and Cyp4a10 expression was increased in the HFD [40,41]. We found that both Cyp3a11 and Cyp4b10 DEGs increased in animals given HFD, whereas these upregulations were attenuated in mice with HFLPD supplementation. The CYP4A14 gene is another one that is significantly induced in HFD-fed animals [42], ob/ob and db/db animals [43,44,45], liver patients, and NAFLD murine models [42]. The expression of CYP4a14 in the livers of HFD-fed animals was elevated, but prevented in HFLPD-fed animals, suggesting its importance in the pathogenesis of nonalcoholic steatohepatitis and simple steatosis [42]. Our findings suggest that the probiotic used in the present study protects the liver from HFD-induced NAFLD by decreasing DEG of CYP4a14. CYP7A1 is a rate-limiting enzyme responsible for converting cholesterol into bile acids in the liver. Treatment with HFD reduced Cyp7a1 mRNA and protein levels in rats [46] and increased its levels in patients with NAFLD [47]. It was found that the DEG of CYP7A1 was significantly induced in the livers of HFD-fed animals, but was downregulated in the livers of HFLPD-fed animals. This was consistent with Jiao et al. [47] and contrary to Wang et al. [46].

Abcc3 is a transporter protein responsible for the basolateral export of anions, including GSH, glucuronide conjugates, and bile salts, from hepatocytes [48]. HFD-fed animals showed reduced Abcc3 gene expression [49]. In contrast to ob/ob and db/db mice, DIO mice exhibited selective induction of Abbc3 and Abbc4 transporters in the liver [50]. Aldh3a2 plays an important role in detoxifying alcohol and lipid peroxidation producing aldehydes. A mutation in Aldh3a2 causes Sjogren–Larsson syndrome [51], involved in lipid droplet formation associated protein [52]. In HFD-fed animals, Aldh3a2 expression increased [53], and it may be a potential drug target for treatment of NAFLD [54]. Likewise, the DEGs of Abcc3 and Aldh3a2 were upregulated in the liver of HFD-fed animals, suggesting that these transcripts may contribute to fat deposition in the liver. HFD-fed mice treated with probiotics had significantly lower DEGs of Abcc3 and Aldh3a2 in the liver, suggesting that the probiotic shows protective effects against diet-induced obesity and its related metabolic liver diseases/disorders.

The Avpr1a protein plays a key role in regulating blood circulation [55], hepatic glucose metabolism, ureagenesis, and fatty acid esterification [56]. A reduction in Avpr1a expression is a key indicator of NAFLD development [57]. The suppression of Avpr1a increases hydrophobic acids in the liver and serum as well as promotes liver inflammation [58]. CD36 induces hepatosteatosis and may contribute to the development of NASH, and several clinical studies have shown that CD36 is closely associated with NAFLD patients and positively correlated with the degree of steatosis in the liver [59]. Based on previous studies, we observed significant increases in AVPR1a and CD36 mRNA expression in the liver of HFD-fed mice, while mice fed with HFLPD showed reduced expression of both transcripts in the liver, suggesting that the supplemented probiotic might play an important role in the regulation of gluconeogenesis and glucose release, as well as inhibiting hepatosteatosis. The enzyme cathepsin D (CTSE) is a lysosomal enzyme and an indicator of NASH [60]. CTSE inhibitors can be regarded as promising and safe NASH drugs [61]. The gene DMBT1 harbors homozygous deletions and/or lacks expression in malignant human brain tumors, and it was named deleted in malignant brain tumors 1 (Dmbt1) [62]. Expression of DMBT1 was associated with inflammation response and liver injury [63]. The DEGs of CTSE and DMBT1 were significantly increased in response to HFD-fed animals, which confirmed that HFD might induce hepatic inflammation and dyslipidemia in mice liver by increasing the mRNA CTSE and DMBT1. However, the supplement with HFLPD inhibited DEGs of CTSE and DMBT1 in mice, suggesting that the probiotic supplement reduced the hepatic inflammation and dyslipidemia via the downregulation of these DEGs.

DBP is a member of the PAR domain basic leucine zipper (PAR bZip) transcription factor family that regulates the enzymes involved in energy metabolism [64]. Inhibition of DBP in 3T3-L1 attenuates PPARγ protein expression during adipogenesis [65]. GAS6 is (profibrogenic factor) one of the main receptors in the liver that has been associated with liver fibrosis [5,66]. Our results revealed that the transcripts of DBP and GAS6 were upregulated in liver of HFD-fed mice, confirming that liver fibrosis and NASH might be developed. However, HFLPD-fed mice showed mRNA expression of both DBP and GAS6 downregulated in liver. H2-Eb1 expression was upregulated in a high-sucrose, high-fat diet [37] and in HFD-fed mice at different time periods [67]. A ductular reaction (DR) is a bile duct hyperplasia accompanied by liver fibrosis, liver injury, and hepatocyte transdifferentiation and regeneration [68]. A trans-fatty acid (TFA)-rich diet promoted the proliferation of bile ducts. The expression of DR indicators and hepatic precursor markers (Krt19, Afp, Epcam, and Cd133 mRNAs) was higher in TFA [69]. In addition, we observed increased expression of Krt19 and H2-Eb1 mRNA in the liver of HFD-fed animals. This may result in liver fibrosis due to a ductular reaction. However, mice fed with HFLPD had reduced expression of Krt19 and H2-Eb1 mRNA in the liver. According to this study, the probiotic appears to contribute to ductular reaction regulation by inhibiting the expression of mRNA Krt19 and H2-Eb1. LY6A induces the expression of interleukin-6 [70]. The DEGs of Ly6a were also significantly upregulated in diabetic mice [71]. MFSD2A is a fasting-inducing gene in the liver that is regulated by both PPAR and glucagon signaling. MFSD2A knockout mice are smaller, leaner, and have reduced serum, liver, and brown adipose triglyceride levels [72]. MOXD1 belongs to the copper-dependent monooxygenase family. It has been found to be upregulated in people with NAFLD [73]. We found higher expression of MFSD2A in the liver of HFD-fed mice, yet the probiotic supplement significantly reduced its expression in mice liver, suggesting that the probiotic supplement would reduce body weight and TG levels of mice liver by reducing MFSD2A mRNA levels. This finding was consistent with body weight and cholesterol levels being reduced as a result of the probiotic treatment. The expression of MOXD1 mRNA in mice treated with probiotics was controversial in the present study. MOXD1 was found to be upregulated in NAFLD [73]. In our study, HFD mice had lower MOXD1 mRNA levels, whereas HFLPD-fed mice had increased levels of MOXD1 mRNA. Myl9 plays a critical role in immune infiltration, tumor invasion, and metastasis. The expression of MYL9 was significantly associated with prognosis in several cancers [74]. MYL9 mRNA expression was higher in mice liver fed with HFD compared to control groups. In mice treated with HFLPD, this expression was reduced; this suggested that our supplemented probiotic might be efficient in protecting the liver from diet-induced obesity and metabolic changes in the liver by inhibiting the carcinogenic marker MYL9. RDH16 enzyme belongs to the short chain dehydrogenase/reductase superfamily, which participates in the metabolism of steroid hormones, prostaglandins, retinoid, and lipids. Xiang et al. found that it was upregulated in db/db mice [37]. Another gene, Saa3, is also increased by acute inflammatory stimuli, and it is linked to obesity. A form of serum amyloid A abundantly expressed in adipose tissue of obese mice is called Saa3 [75]. Tceal8 was positively correlated with glucose intolerance of white blood cells [76] and upregulated in HSHF-induced NAFLD and db/db mice [37]. Upp2 is a liver-specific protein that is essential for pyrimidine salvage reactions [77,78]. Upp2 inhibition reduced the level of endogenous uridine in the liver, which protects the liver from drug-induced lipid accumulation [77,79]. It was found that the HFD supplement induced a sharp increase in the expression of RDH16, Saa3, and Upp2 mRNA in mice livers compared to the control group. The expression of both mRNA transcripts was significantly downregulated in animals fed with HFLPD. It has been reported that *L. plantarum* plays a major role in reducing fat mass and its size through regulating genes associated with lipogenesis and fatty acid oxidation via modulating microbiota in GIT. Based on our previous study, we found that *L. plantarum*-A29 (KCC-48) could survive in GIT of diet-induced obese mice, which was confirmed by pyrosequencing. In addition, it reduces adipose tissue mass by downregulating key transcription factors and downstream targets associated with lipid synthesis via pathways including p38MAPK, P44/42, and AMPK-α [10]. The number of scientific reports on the effects of probiotic *L. plantarum* on global gene expression in obese animals is still limited. Our knowledge is that this is the first report describing the transcriptome changes in liver tissues of obese animals fed with *L. plantarum*-KCC-48. The majority of the study’s findings are in agreement with the transcriptional changes observed in diet-induced obesity, with the exception of a few studies that contradict ours. Further research will be necessary to re-validate the reported mRNA and their protein expressions using PCR and immunotechniques, respectively.

## 4. Materials and Methods

### 4.1. Diet and Lactiplantibacillus plantarum-KCC48

We obtained normal (AN93) and high-fat diets (45% fat calories) from Feed Korea lab diets in South Korea. The strain *L**. plantarum*-KCC48, which has been previously isolated and characterized [10], was grown in MRS broth at 37 ± 2 °C for 24 h, after which the pellets were harvested by centrifugation at 4 °C for 30 min. Pellets were washed twice with phosphate-buffered saline. The pellets of *L. plantarum*-KCC48 were mixed with a high-fat diet (Feed Korea Lab diet) and considered a HFLPD diet. The final concentration of *L. plantarum*-KCC48 was 10^9^ CFU/4 g of high-fat diet.

### 4.2. Animals and Probiotic Diet Production

Male ICR mice (25 ± 3 g/seven weeks old) were obtained from Orient Bio (Seongnam, South Korea). This study was carried out in accordance with the procedures in the Animal Research: Reporting of in-vivo Experiments (ARRIVE) manual and the recommendations in the Guide for Animal Care and Use at Chonbuk National University (Jeollabuk-do, Korea). The experimental design and the procedure were approved by the University Committee on Ethics in the Care and Use of Laboratory Animals. Compositions and energy content of the normal diet and the high-fat diet are given in the Appendix A. Animals were kept in air-conditioned rooms with a 12 h light/dark cycle at 20 °C and 2 °C, respectively. For a week, mice were allowed free access to water and normal food in the experimental facility.

### 4.3. Experimental Design

A total of 30 mice were divided into three groups of 10 each. Animals in Group I were fed a normal diet; animals in Group III were fed a high-fat diet with *L. plantarum* (10^9^ CFU/animal; HFLPD) for eight weeks (Figure 4). Every morning at 9 a.m., HFD and HFLPD diets were supplemented to the respective groups. Diets and water intake were regularly monitored. Padding material was changed twice a week. The total experimentation period was 8 weeks. At the end of the experimentation period, all mice were anesthetized using isoflurane and were sacrificed after 12 h of fasting. The liver tissues of each experimental animal were weighed immediately and then stored at −80 °C for further analysis.

### 4.4. Liver Triglyceride Quantification

Liver tissue (100 mg) from each animal was collected and washed twice with ice-cold PBS, then homogenized under liquid nitrogen and reconstituted in double-distilled H_2_O with 5% Np-40. Then, we slowly heated the samples to 80–100 °C in a water bath for 2–5 min, or until the NP-40 solution became cloudy, then cooled down to room temperature and centrifuged for 2 min at maximum speed to remove the insoluble materials. The triglyceride content of each sample was determined using a TG assay kit (Abcam, Cambridge, MA, USA).

### 4.5. RNA Extraction

The total RNA was extracted from liver tissues of experimental samples using Trizol reagent and RNA lipid tissue mini kit (Qiagen, Valencia, CA, USA) and the quality was determined by an Agilent 2100 bio-analyzer (Agilent technologies, Amstelveen, Netherlands). RNA was quantified using an ND-2000 spectrophotometer (Thermo Inc., Waltham, MA, USA).

### 4.6. Library Preparation and Sequencing

The NEB Next Ultra II Directional RNA-Seq Kit (New England BioLabs, Inc., Ipswich, MA, USA) was used to construct total RNA libraries. The mRNA was then isolated using a Poly (A) RNA Selection Kit (Lexogen Inc., Vienna, Austria). Complementary DNA (cDNA) was then synthesized from isolated mRNA and sheared according to the manufacture protocols. Indexing was performed by the Illumina indexes 1–12. PCR was used to enrich the samples. The libraries were then screened with a TapeStation HS D1000 Screen Tape (Agilent Technologies, Amstelveen, The Netherlands) to determine fragment size and quantified with a StepOne Real-Time PCR System (Life Technologies Inc., Carlsbad, CA, USA). Sequencing for high throughput was performed with NovaSeq 6000 (Illumina Inc., San Diego, CA, USA).

### 4.7. Data Analysis and Removal of Low Quality Reads

Raw sequence quality control was carried out by (https://www.bioinformatics.babraham.ac.uk/projects/fastqc/, accessed on 1 April 2022), and adapter and low-quality reads were removed by FASTX_Trimmer (FASTX toolkit; FASTX-Toolkit (cshl.edu)) and BBMap (BBMap download|SourceForge.net). TopHat was used to map the quality reads to the reference genome [80]. Read count data were processed using EdgeR’s FPKM + Geometric normalization method in R [81]. The fragments per kb per million reads (FPKM) were calculated using Cufflinks [82]. The data mining and graphic visualization were performed with ExDEGA (Ebiogen Inc., Seoul, Korea). DAVID and ExDEGA graphicPlus were used to analyze the functional annotation and KEGC pathways (http://david.abcc.ncifcrf.gov/, accessed on 1 April 2022).

## 5. Conclusions

mRNA sequencing was used to study the response of *L. plantarum*-KCC48 to an HFD-induced fatty liver disease model in mice. Based on the data derived from the present study, HFLPD diet treatment to HFD-fed mice led to a significant reduction in liver weight and a normalization of hepatic triglyceride levels. Based on liver transcriptome data, we found that 72.41% of differentially expressed genes (DEGs) were upregulated and 27.59% of DEGs were downregulated in the liver tissues of animals on HFD diets. Animals fed the HFLPD showed that 59.02% of DEGs were upregulated and 41.98% of DEGs were downregulated compared to HFD-fed animals. According to the study, probiotic *L. plantarum* treatment of the liver regulates the HFD-induced transcriptome changes that are closely associated with fatty liver disease, protecting the liver from HFD-induced abnormalities as well as metabolic changes. We have demonstrated that *L. plantarum*-KCC48 is a novel probiotic that may be desirable for the prevention of diet-induced fatty liver diseases.

## Figures and Tables

**Figure 1 ijms-23-06750-f001:**
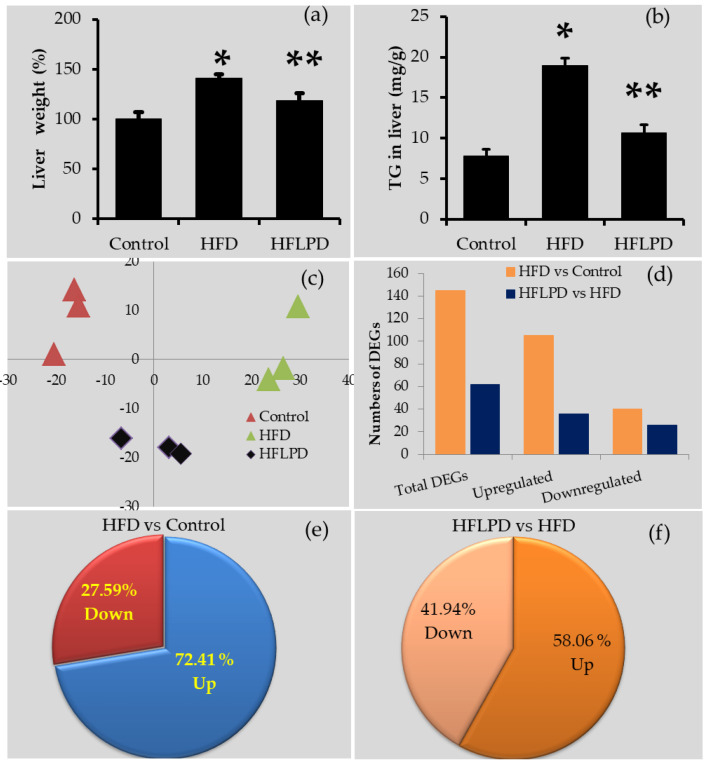
The total liver weight, triglyceride content, and the number of differentially expressed genes (DEGs) in the liver of experimental animals. (**a**) Total weight of liver tissues of experimental animals, (**b**) triglyceride content of liver tissues of experimental animals. These data are represented by the mean ± standard deviation, total liver weight (*n* = 5), and TG content (*n* = 3). * *p* < 0.05 HFD vs. Control; ** *p* < 0.05 HFLPD vs. HFD. (**c**) Principal component analysis (PCA) of each group, (**d**) total number of DEGs in the liver tissues of HFD- and HFLPD-fed mice (*p* < 0.05, greater than 2-fold), (**e**) percentage of up- and downregulated DEGs in the liver tissue of HFD vs. Control, (**f**) percentage of up and downregulated DEGs in the liver tissue of HFLPD vs. HFD.

**Figure 2 ijms-23-06750-f002:**
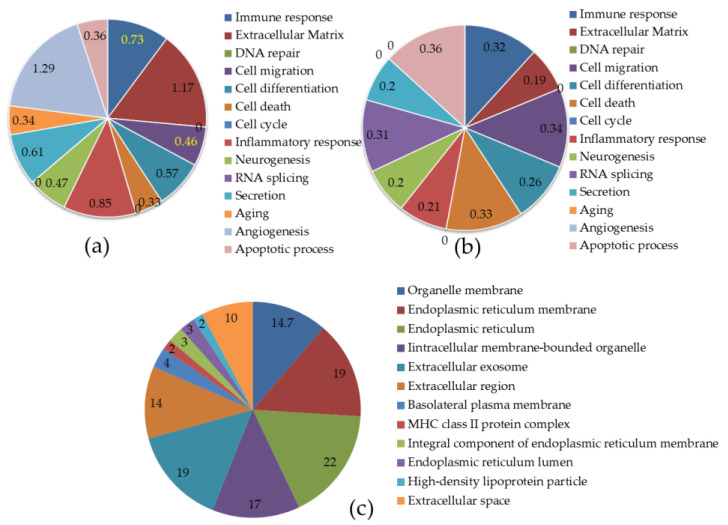
Gene categories and localization of DEGs detected in the liver of experimental animals. (**a**) Gene categorization of identified DEGs in the liver of HFD-fed mice compared to control mice (*p* < 0.05, greater than 2-fold); (**b**) Gene categorization of identified DEGs in the liver of HFLPD-fed mice compared to HFD (*p* < 0.05, greater than 2-fold); (**c**) distribution of identified DEGs in the liver of experimental animals.

**Figure 3 ijms-23-06750-f003:**
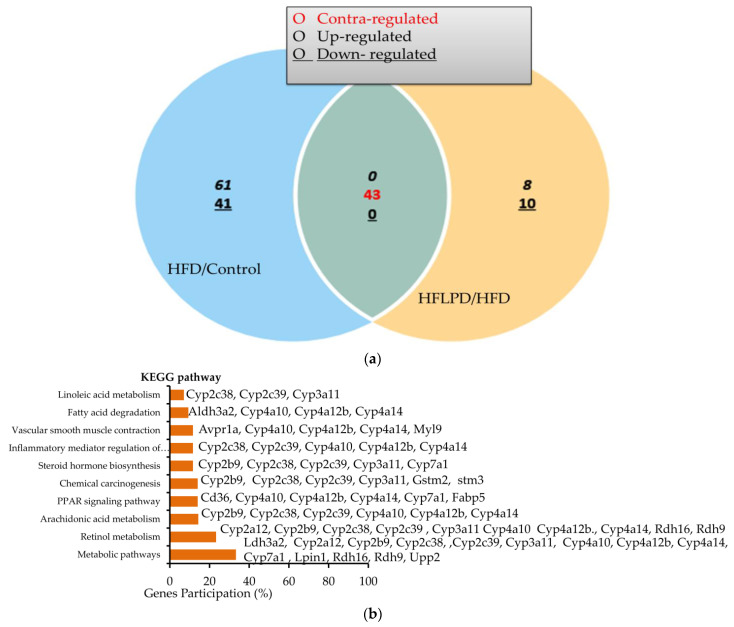
Contraregulation of identified DEGs in the liver tissue between HFD- and HFLPD-fed animals. In total, 43 DEGs were identified in both HFD- and HFLPD-fed animals that were contraregulated. (**a**) Venn diagram of the DEGs identified among HFD- and HFLPD-fed animals. (**b**) KEGG pathway of the DEGs. (**c**) Heat map views of the DEGs detected between HFD and HFLPD animals.

**Figure 4 ijms-23-06750-f004:**
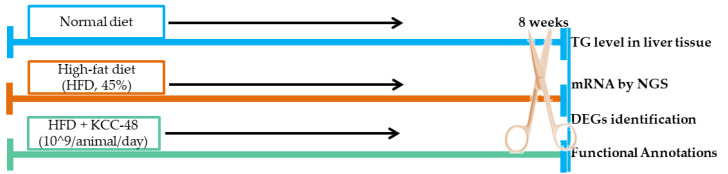
Experimental design and diet intervention. TG, triglycerides; NGS, Next-Generation Sequencing; DEGs, differentially expressed genes.

**Table 1 ijms-23-06750-t001:** Contraregulation of DEGs in the liver of HFD- and HFLPD-fed mice.

S. No	Gene Symbol	Gene Name	Fold Changes	
			HFD/Control	HFLPD/HFD
1	9030619P08Rik	lymphocyte antigen 6 complex pseudogene (9030619P08Rik)	2.094	0.493
2	Abcc3	ATP-binding cassette, sub-family C (CFTR/MRP), member 3 (Abcc3)	3.739	0.373
3	Aldh3a2	aldehyde dehydrogenase family 3, subfamily A2 (Aldh3a2)	6.414	0.448
4	Avpr1a	arginine vasopressin receptor 1A (Avpr1a)	0.454	2.216
5	Cd36	CD36 antigen (Cd36)	3.558	0.375
6	Ces1g	carboxylesterase 1G (Ces1g)	2.228	0.356
7	Ctse	cathepsin E (Ctse)	3.918	0.382
8	Cyp2a12	cytochrome P450, family 2, subfamily a, polypeptide 12 (Cyp2a12)	0.427	2.902
9	Cyp2b9	cytochrome P450, family 2, subfamily b, polypeptide 9 (Cyp2b9)	13.813	0.351
10	Cyp2c38	cytochrome P450, family 2, subfamily c, polypeptide 38 (Cyp2c38)	6.591	0.273
11	Cyp2c39	cytochrome P450, family 2, subfamily c, polypeptide 39 (Cyp2c39)	9.718	0.243
12	Cyp3a11	cytochrome P450, family 3, subfamily a, polypeptide 11 (Cyp3a11)	5.499	0.460
13	Cyp3a59	cytochrome P450, family 3, subfamily a, polypeptide 59 (Cyp3a59)	4.626	0.397
14	Cyp4a10	cytochrome P450, family 4, subfamily a, polypeptide 10 (Cyp4a10)	26.691	0.186
15	Cyp4a12b	cytochrome P450, family 4, subfamily a, polypeptide 12B (Cyp4a12b)	2.282	0.456
16	Cyp4a14	cytochrome P450, family 4, subfamily a, polypeptide 14 (Cyp4a14)	24.394	0.180
17	Cyp7a1	cytochrome P450, family 7, subfamily a, polypeptide 1 (Cyp7a1)	4.482	0.454
18	Dbp	D site albumin promoter binding protein (Dbp)	13.345	0.355
19	Dmbt1	deleted in malignant brain tumors 1 (Dmbt1)	47.186	0.024
20	Elovl3	elongation of very long chain fatty acids	0.271	2.786
21	Fabp5	fatty acid binding protein 5, epidermal (Fabp5)	0.076	2.275
22	Fam25c	family with sequence similarity 25, member C (Fam25c)	0.168	9.567
23	Gadd45g	growth arrest and DNA-damage-inducible 45 gamma (Gadd45g)	0.267	3.741
24	Gas6	growth arrest specific 6 (Gas6)	2.145	0.455
25	Gm3219	B-cell CLL/lymphoma 7C pseudogene (Gm3219)	0.401	3.253
26	Gstm2	glutathione S-transferase, mu 2 (Gstm2)	2.275	0.430
27	Gstm3	glutathione S-transferase, mu 3 (Gstm3)	3.361	0.249
28	H2-Eb1	histocompatibility 2, class II antigen E beta (H2-Eb1)	3.240	0.361
29	Krt19	keratin 19 (Krt19)	4.125	0.287
30	Lpin1	lipin 1 (Lpin1)	2.546	0.403
31	Ly6a	lymphocyte antigen 6 complex, locus A (Ly6a)	8.768	0.172
32	Ly6c1	lymphocyte antigen 6 complex, locus C1 (Ly6c1)	8.054	0.166
33	Ly6d	lymphocyte antigen 6 complex, locus D (Ly6d)	5.567	0.463
34	Mfsd2a	major facilitator superfamily domain containing 2A (Mfsd2a)	4.328	0.352
35	Moxd1	monooxygenase, DBH-like 1 (Moxd1)	0.020	31.917
36	Mup21	major urinary protein 21 (Mup21)	0.456	2.180
37	Myl9	myosin, light polypeptide 9, regulatory (Myl9)	2.474	0.405
38	Rdh16	retinol dehydrogenase 16 (Rdh16)	3.376	0.283
39	Rdh9	retinol dehydrogenase 9 (Rdh9)	3.309	0.350
40	Saa3	serum amyloid A 3 (Saa3)	3.576	0.448
41	Slc16a7	solute carrier family 16 (monocarboxylic acid transporters), member 7	2.132	0.487
42	Tceal8	transcription elongation factor A (SII)-like 8 (Tceal8)	2.259	0.420
43	Upp2	uridine phosphorylase 2 (Upp2)	5.374	0.443

**Table 2 ijms-23-06750-t002:** Functional annotations of differentially expressed genes in HFD-fed animals compared to control animals.

S. No.	Term	Count	%	*p*-Value
1.	Lipid metabolism	21	14.5	9 × 10^−13^
2.	Cholesterol metabolism	11	7.6	1.1 × 10^−12^
3.	Steroid metabolism	11	7.6	2.4 × 10^−11^
4.	Cholesterol metabolic process	12	8.3	3.3 × 10^−11^
5.	Fatty acid metabolic process	11	7.6	1.6 × 10^−7^
6.	Lipid biosynthesis	12	8.3	3.8 × 10^−9^
7.	Lipid metabolic process	23	15.9	1.2 × 10^−12^
8.	Steroid metabolic process	10	6.9	7.9 × 10^−9^
9.	Steroid hormone biosynthesis	10	6.9	1.9 × 10^−7^
10.	Endoplasmic reticulum	32	22.1	3.3 × 10^−14^
11.	Organelle membrane	14	9.7	5.9 × 10^−14^
12.	Microsome	15	10.3	6.5 × 10^−14^
13.	Heme binding	17	11.7	6.6 × 10^−14^
14.	Endoplasmic reticulum membrane	28	19.3	1.9 × 10^−13^
15.	Metal ion binding site:iron (heme axial ligand)	14	9.7	8.9 × 10^−13^
16.	Iron ion binding	17	11.7	1.3 × 10^−12^
17.	Secondary metabolites biosynthesis, transport, and catabolism	17	11.7	2.5 × 10^−12^
18.	Sterol metabolism	11	7.6	3.8 × 10^−12^
19.	Cytochrome p450, *e*-class, group i	12	8.3	5.8 × 10^−12^
20.	Endoplasmic reticulum	35	24.1	6 × 10^−12^
21.	Metabolic pathways	40	27.6	6.5 × 10^−12^
22.	Iron	19	13.1	1.3 × 10^−11^
23.	PPAR signaling pathway	13	9	1.7 × 10^−11^
24.	Intracellular membrane-bounded organelle	25	17.2	1.8 × 10^−10^
25.	Chemical carcinogenesis	12	8.3	1.6 × 10^−9^
26.	Arachidonic acid metabolism	10	6.9	2.3 × 10^−7^
27.	Disulfide bond	40	27.6	5.1 × 10^−6^
28.	Metabolic process	14	9.7	0.000025
29.	Biosynthesis of antibiotics	11	7.6	0.000054
30.	Signal	47	32.4	0.00013
31.	Extracellular exosome	33	22.8	0.00068
32.	Glycoprotein	39	26.9	0.00091
33.	Extracellular space	22	15.2	0.001
34.	Secreted	22	15.2	0.0012
35.	Disulfide bond	31	21.4	0.0013
36.	Extracellular region	24	16.6	0.0014
37.	Acetylation	33	22.8	0.0017
38.	Catalytic activity	11	7.6	0.0017
39.	Signal peptide	35	24.1	0.0029
40.	Protein homodimerization activity	13	9	0.0089
41.	Membrane	59	40.7	0.029
42.	Metal binding	30	20.7	0.032
43.	Lipoprotein	10	6.9	0.047
44.	Hydrolase activity	17	11.7	0.061

**Table 3 ijms-23-06750-t003:** Functional annotations of differentially expressed genes in HFLPD-fed animals compared to HFD-fed animals.

S. No.	Term	Count	%	*p*-Value
1.	Monooxygenase	13	21.7	5.50 × 10^−16^
2.	Cytochrome p450, conserved site	12	20	1.20 × 10^−15^
3.	Cytochrome p450	12	20	3.10 × 10^−15^
4.	Retinol metabolism	12	20	1.90 × 10^−14^
5.	Oxidoreductase activity, acting on paired donors	11	18.3	2.30 × 10^−13^
6.	Heme	12	20	4.40 × 10^−13^
7.	Iron ion binding	13	21.7	7.90 × 10^−13^
8.	Heme binding	12	20	2.50 × 10^−12^
9.	Organelle membrane	10	16.7	3.90 × 10^−12^
10.	Monooxygenase activity	10	16.7	2.30 × 10^−11^
11.	Secondary metabolites biosynthesis, transport	12	20	3.10 × 10^−11^
12.	Metal ion binding site:iron (heme axial ligand)	10	16.7	3.50 × 10^−11^
13.	Microsome	10	16.7	4.10 × 10^−11^
14.	Iron	13	21.7	1.30 × 10^−10^
15.	Oxidoreductase	15	25	4.30 × 10^−10^
16.	Endoplasmic reticulum	17	28.3	1.80 × 10^−9^
17.	Endoplasmic reticulum membrane	15	25	4.80 × 10^−9^
18.	Intracellular membrane-bounded organelle	13	21.7	6.90 × 10^−7^
19.	Endoplasmic reticulum	16	26.7	1.70 × 10^−6^
20.	Oxidation-reduction process	12	20	2.20 × 10^−6^
21.	Metabolic pathways	17	28.3	6.90 × 10^−6^
22.	Oxidoreductase activity	11	18.3	8.10 × 10^−6^
23.	Metal binding	17	28.3	7.90 × 10^−3^
24.	Disulfide bond	16	26.7	8.80 × 10^−3^
25.	Glycoprotein	18	30	1.10 × 10^−2^
26.	Signal	19	31.7	2.60 × 10^−2^
27.	Disulfide bond	13	21.7	5.10 × 10^−2^
28.	Membrane	29	48.3	6.20 × 10^−2^

**Table 4 ijms-23-06750-t004:** KEGG signaling enrichment analysis between HFD-fed animals vs. control animals by the DAVID Bioinformatics tool.

	Term	Counts	%	*p*-Value
1.	Retinol metabolism	18	12.4	8.4 × 10^−18^
2.	Metabolic pathways	40	27.6	6.5 × 10^−12^
3.	PPAR signaling pathway	13	9	1.7 × 10^−11^
4.	Chemical carcinogenesis	12	8.3	1.6 × 10^−9^
5.	Steroid hormone biosynthesis	10	6.9	0.00000019
6.	Arachidonic acid metabolism	10	6.9	0.00000023
7.	Fatty acid degradation	7	4.8	0.0000089
8.	Linoleic acid metabolism	7	4.8	0.00001
9.	Inflammatory mediator regulation of TRP channels	9	6.2	0.000037
10.	Biosynthesis of antibiotics	11	7.6	0.000054
11.	Terpenoid backbone biosynthesis	4	2.8	0.0015
12.	Propanoate metabolism	4	2.8	0.0025
13.	Serotonergic synapse	6	4.1	0.011
14.	Fatty acid metabolism	4	2.8	0.015
15.	Steroid biosynthesis	3	2.1	0.016
16.	Valine, leucine, and isoleucine degradation	4	2.8	0.018
17.	Asthma	3	2.1	0.024
18.	Metabolism of xenobiotics by cytochrome P450	4	2.8	0.027
19.	Circadian rhythm	3	2.1	0.039
20.	Vascular smooth muscle contraction	5	3.4	0.04
21.	beta-alanine metabolism	3	2.1	0.044
22.	Antigen processing and presentation	4	2.8	0.05
23.	Intestinal immune network for IgA production	3	2.1	0.067
24.	Tryptophan metabolism	3	2.1	0.082
25.	*Staphylococcus aureus* infection	3	2.1	0.091
26.	Drug metabolism—other enzymes	3	2.1	0.094
27.	Graft-versus-host disease	3	2.1	0.097

**Table 5 ijms-23-06750-t005:** KEGG signaling enrichment analysis between HFLPD-fed animals vs. HFD-fed animals by the DAVID Bioinformatics tool.

	Term	Count	%	*p*-Value
1.	Retinol metabolism	12	20	1.90 × 10^−14^
2.	PPAR signaling pathway	8	13.3	2.20 × 10^−8^
3.	Chemical carcinogenesis	8	13.3	5.80 × 10^−8^
4.	Arachidonic acid metabolism	7	11.7	1.20 × 10^−6^
5.	Metabolic pathways	17	28.3	6.90 × 10^−6^
6.	Steroid hormone biosynthesis	6	10	2.20 × 10^−5^
7.	Fatty acid degradation	4	6.7	9.60 × 10^−4^
8.	Inflammatory mediator regulation of TRP channels	5	8.3	1.50 × 10^−3^
9.	Vascular smooth muscle contraction	5	8.3	1.60 × 10^−3^
10.	Linoleic acid metabolism	3	5	1.70 × 10^−2^

## Data Availability

The experimental data are available on request from the corresponding author.

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
