# Peer review of "A Transcriptomic Response to Lactiplantibacillus plantarum-KCC48 against High-Fat Diet-Induced Fatty Liver Diseases in Mice"

_ijms, 2022, doi:10.3390/ijms23126750_

Round 1

Reviewer 1 Report

In the present work, the authors evaluated the probiotic activity of L. plantarum KCC48 for the prevention of fatty liver disease induced by a high-fat diet in mice.

Next-generation sequencing technologies were used to examine differential gene expression between mice fed a high-fat diet and those fed a high-fat diet along with L. plantarum KCC48.

The topic of the study is interesting and in accordance with the current research objectives on the use of probiotics as anti-obesity agents. The experimental design is appropriate and the content of the work is noteworthy. However, the manuscript cannot be accepted in its current form because there are some aspects that the authors should clarify and improve.

The manuscript has gaps in the materials and methods section and in the section on the presentation of results.

In detail:

Materials and methods:

Par. 4.2

-       Table of animal feed is not reported.

Par. 4.3

-        Please check and improve experimental design description

The enrichment analysis should be better described

Statistical analysis methods are not reported

Results section:

Par 2.1

-       Figure 1 captions does not correspond to figure. Please check.

-       What statistical analysis was carried out? Please insert.

-       Who is HFD29 in figure 1d?

Par. 2.2

-       Lines 138-139. Check the sentence and the respective pie chart in Figure 1f. There is discordance between genes up- and down-regulated.

Par. 2.3

-       GO terms and values in the text do not correspond to those shown in Figure 2c. Please check.

Par 2.4

-       The number of up- and down-regulated genes does not correspond to that shown in table 1. Please check.

-       HFDA29 is shown in Figures 3b and 3c. Perhaps you meant KCC48. Please check.

-       Figure 3b is redundant with Table 1. If not strictly necessary, it can also be eliminated or shown as supplementary material.

Check the name of the strain. Sometimes it is reported as KCC48 others as A29

The authors should use the new taxonomy of the genus Lactobacillus (https://doi.org/10.1099/ijsem.0.004107)

Author Response

Reviewer comments 1

In the present work, the authors evaluated the probiotic activity of L. plantarum KCC48 for the prevention of fatty liver disease induced by a high-fat diet in mice. Next-generation sequencing technologies were used to examine differential gene expression between mice fed a high-fat diet and those fed a high-fat diet along with L. plantarum KCC48. The topic of the study is interesting and in accordance with the current research objectives on the use of probiotics as anti-obesity agents. The experimental design is appropriate and the content of the work is noteworthy. However, the manuscript cannot be accepted in its current form because there are some aspects that the authors should clarify and improve. The manuscript has gaps in the materials and methods section and in the section on the presentation of results.

I would like to thank the reviewer for their positive comments on the research article submitted as well as their suggestions that will help to improve the quality of the manuscript. A problem arises between sections on materials and methods and results, in this study we carried out an invivo mouse experimental analysis to assess the efficacy of the probiotic L plantarum-KCC48 against high fat diet induced fat depositions that lead to liver transcriptome changes. The majority of the analyses were performed by ebiogen, Seoul, Korea. We analyzed differentially expressed genes, as well as their functional annotations and others using several online tools presented in the materials and methods section of this paper. Due to this, the experimental protocols have been abbreviated in this manuscript, and the results section has been elaborated.  In response to the reviewers' comments, we have carefully read the entire manuscript and modified it in accordance with their suggestions. Please note that the changes have been highlighted in red throughout this document.

Materials and methods:

  1. 4.2Table of animal feed is not reported.

Yes, we are in agreement with the reviewer's comment.  Detailed information regarding feed compositions and their energy contents can now be found in accordance with supplementary table 3Par.

  1. 3-        Please check and improve experimental design description

Thank you, we have significantly improved the experimental protocol compared to the original submission. The purpose of this study was to determine the efficacy of probiotic L plantarum- KCC48 against high fat diet induced fat depositions and transcriptome changes in the liver. The majority of the analyses were carried out by ebiogen, Seoul, Korea. In the next step, we analyzed differentially expressed genes, their functional annotations, as well as several other tools that were presented in the section on materials and methods. 

  1. The enrichment analysis should be better described

The enrichment data analysis significantly described in the result and discussion section (Line Nos)

  1. Statistical analysis methods are not reported

Differentially expressed genes and their statistical analysis were performed by ebiogen, Seoul, Korea using R software that was cited in the method section. Read count data were processed using EdgeR's FPKM+Geometric normalization method in R (R Developmen Core Team. A Language and Environment for Statistical Computing. In Foundation for Statistical Computing, Vienna, 2020). Further, the data obtained from ebiogen, Seoul, Korea were subjected into functional annotation clustering, chart, table and gene classification using online tool, DAVID Bioinformatics Resources (DAVID: Functional Annotation Tools (ncifcrf.gov))

  1. Par 2.1-   Figure 1 captions do not correspond to figure. Please check.

Yes, we have agreed with the reviewer's comments.  All the mistakes have now been corrected.

Figrure1. The total liver weight, triglyceride content, and the number of differentially expressed genes (DEGs) in the liver of experimental animals. (a) Total weight of liver tissues of experimental animals, (b) triglyceride content of liver tissues of experimental animals. These data are represented by the mean ± standard deviation, total liver weight (n=5, p<0.05) and TG content (n=3, p<0.05). The different alphabets within figures indicate a significant difference between experimental groups, (c) principal components analysis (PCA) of each group, (d) total number of DEGs in the liver tissues of HFD and HFLPD fed mice (p<0.05, greater than 2 fold), (e) percentage of up and down regulated DEGs in the liver tissue of HFD vs Control, (f) percentage of up and down regulated DEGs in the liver tissue of HFLPD vs HFD

  1. -       What statistical analysis was carried out? Please insert.

Differentially expressed genes and their statistical analysis were performed by ebiogen, Seoul, Korea using R software that was cited in the method section. Read count data were processed using EdgeR's FPKM+Geometric normalization method in R (R Developmen Core Team. A Language and Environment for Statistical Computing. In Foundation for Statistical Computing, Vienna, 2020). Further, the data obtained from ebiogen, Seoul, Korea were subjected into functional annotation clustering, chart, table and gene classification using online tool, DAVID Bioinformatics Resources (DAVID: Functional Annotation Tools (ncifcrf.gov))

  1. -       Who is HFD29 in figure 1d?

Thank you for pointing out the typographical error. It has now been corrected to read exactly as HFLPD 

  1. -       Lines 138-139. Check the sentence and the respective pie chart in Figure 1f. There is discordance between genes up- and down-regulated.

Yes, we have agreed with the comment of the reviewer. There was a mistake in figure 1f in the previous version. It has been revised and interpreted correctly according to the data presented in the figure.

Animal with HFD had 145 differentially expressed genes in liver tissue compared to animal fed with a normal diet. On the basis of the total number of genes expressed, 105 genes (72.41%) were upregulated while 40 genes (27.59%) were downregulated (p<0.05) as compared with animals from the control group (Figure 1d & e) and see Supplementary Table 1).  However, animal fed with HFLPD were found to have 61 differentially expressed genes (p<0.05) in the liver tissue of animals compared to animals fed with HFD. There were 36 genes (59.02%) upregulated and 25 genes (40.98%) downregulated (Figure 1d & f and see the supplementary table 2).

  1. -       GO terms and values in the text do not correspond to those shown in Figure 2c. Please check.

 There is a mistake in the text regarding the GO terms. We have corrected it in figure 2c  

  1. -       The number of up- and down-regulated genes does not correspond to that shown in table 1. Please check.

Yes, now it has revised as a total 43 transcripts were identified in liver tissues between HFD and HFLPD that were associated with contraregulation of biological process, Of these, 34 genes were significantly upregulated and 9 genes were downregulated in HFD fed mice, while 34 genes were downregulated and 9 were upregulated in response to HFLPD treatment. 

  1. HFDA29 is shown in Figures 3b and 3c. Perhaps you meant KCC48. Please check.

Thank you very much for the information you provided. In the whole manuscript, the abbreviation for HFDA29 has been uniformly changed from HFDA29 to HFLPD..

  1. -       Figure 3b is redundant with Table 1. If not strictly necessary, it can also be eliminated or shown as supplementary material.

We agree with the reviewer's comments and have removed figure 3b from figure 3 as a result

  1. Check the name of the strain. Sometimes it is reported as KCC48 others as A29

Thank you for providing this information. As a result of these revisions, the abbreviation for HFDA29 has been uniformly abbreviated throughout the manuscript as HFLPD.

  1. The authors should use the new taxonomy of the genus Lactobacillus (https://doi.org/10.1099/ijsem.0.004107)

I would like to thank you for your information. The new taxonomy name for Lactobacillus plantarum has been revised and references have been added.

Reviewer 2 Report

The authors of paper titled:" A Transcriptomic Response to Lactobacillus plantarum-KCC48 against High-Fat Diet Induced Fatty Liver Diseases In Mice", described an analysis of the differential expression of genes involved in adipogenesis and the effect of Lactobacillus Plantarum on the development of fatty liver in mice.

The work has been done well, the results are clear, and the article is easy to read.

However, there are some technical errors in the manuscript:

Line 106 – “tin” chang to “in”

Line 169 –“retinal” to retinol

Line 232 – “lipid metabolism” wrote twice

Line 288-289 - Please enter the reference number for  Jiao et al. and Wang et al.

Line 396 – 397 – “Nutrient content of animal feed is shown in Table (Table)”. I have not seen Table.

Line 401-406 – “30 mice were divided into two groups of 10 each” – may be three groups? You missed Group II (HFD)

 Please discuss article by Takemura N. et al. (Exp Biol Med (Maywood). 2010 Jul;235(7):849-56).

Author Response

Reviewer 2

The authors of paper titled:" A Transcriptomic Response to Lactobacillus plantarum-KCC48 against High-Fat Diet Induced Fatty Liver Diseases In Mice", described an analysis of the differential expression of genes involved in adipogenesis and the effect of Lactobacillus Plantarum on the development of fatty liver in mice. The work has been done well, the results are clear, and the article is easy to read. However, there are some technical errors in the manuscript:

The author wishes to thank the reviewers for their positive comments on the submitted research paper and suggestions that would help to improve the quality of the manuscript. As a response to the reviewers' suggestions, we have read through the entire manuscript and modified it in accordance with their suggestions. Please note that the changes have been marked in red throughout the manuscript.

  1. Line 106 – “tin” chang to “in”

Thanks for pointing this out. “It is interesting to note that this finding was strongly supported by the results of the weights of liver in experimental animals

  1. Line 169 –“retinal” to retinol

Thank you, this mistake has been corrected in the abstract and the results section.

  1. Line 232 – “lipid metabolism” wrote twice

Yes, the repetition in the manuscript has been removed 

  1. Line 288-289 - Please enter the reference number for Jiao et al. and Wang et al.

Yes, we have included and cited the references for  Jiao et al. and Wang et al ( Reference Nos : 45 and 46

  1. Line 396 – 397 – “Nutrient content of animal feed is shown in Table (Table)”. I have not seen Table.

Yes, we are in agreement with the reviewer's comment.  Detailed information regarding feed compositions and their energy contents can now be found in accordance with supplementary table 3

  1. Line 401-406 – “30 mice were divided into two groups of 10 each” – may be three groups? You missed Group II (HFD)

Thank you for your essential information. A total of 30 mice were divided into three groups of 10 each. Animals in Group I was fed a normal diet; animals in Group III were fed a high fat diet with L. plantarum (106 CFU/animal; HFLPD) for eight weeks.

  1. Please discuss article by Takemura N. et al. (Exp Biol Med (Maywood). 2010 Jul; 235(7):849-56).

This reference has been used in the introduction section of the paper. L plantarum reduced fat percentages in healthy volunteers and decreased the size of adipocytes in mice. Furthermore, it reduced diet-induced obesity by reducing adipocyte size (Takemura et al., 2010)

Reviewer 3 Report

Manuscript # 1745422

In the current manuscript titled “ A Transcriptomic Response to Lactiplantibacillus plantarum-KCC48  against High-Fat Diet-Induced Fatty Liver Diseases in Mice” the authors have determined the transcriptome analysis to establish the correlation between gut microbes (microbiota) and its alteration (dysbiosis) in the progression of high-fat diet-induced Fatty Liver disease. Authors have focused on the role of the specific bacterium Lactiplantibacillus plantarum-KCC48 in the regulation of fat metabolism and regulation of obesity and related fatty liver disease including non-alcoholic fatty liver disease (NAFLD). This is an interesting piece of transcriptome analysis work that will strengthen our understanding in the field of microbiome and related metabolic diseases. However, I have the following comments for this study:

1)    Although authors have given an explanation that the beneficial bacterium Lactiplantibacillus plantarum-KCC48 is involved in the reduction of fat percentage and adipocytes numbers. Since obesity involves an increase in adipocyte number and size. Thus, did the authors find any explanation regarding the role of Lactiplantibacillus plantarum-KCC48 in the reduction of adipocytes number and size both?

2)    How is this bacterium involved in the reduction of inhibits inflammation, dyslipidemia, hypocholesterolemia, insulin resistance, and obesity, as well as modulating gut microbiota? Do authors want to say that these changes in metabolic pathways are related to gut dysbiosis? ( https://pubmed.ncbi.nlm.nih.gov/26999104/, ).  How does this bacterium affect liver metabolism? Authors can give their supporting explanation if it is available?

3)     Why we are blaming the fat alone for obesity. Current studies ate emphasize the role of sugar,  carbohydrates and saturated fatty acids in Fatty liver disease (https://pubmed.ncbi.nlm.nih.gov/34257427/, https://www.sciencedirect.com/science/article/pii/S1551714417306018, https://pubmed.ncbi.nlm.nih.gov/33923255/). Authors should talk about carbohydrates metabolism too in liver disease.

4)    How is this bacterium regulating adipose tissue inflammation? Authors can give their explanation through literature support. In their transcriptome analysis, NLRP12 was increased in the HFD condition as compared to control. What is the status in supplemented group?

5)    In figure 1 (Figure 1a and 1b)  there is an annotation on the top of the bar c, a b. The author can represent this figure with * and p values. It is not explained in the figure legend.

6)    The title of the main manuscript and Supplementary figure is different. The authors should correct it.

7)    In lines 402-403, experimental design section, the authors have written that 30 mice were taken and divided into two groups of 10 mice. I did not understand the groups. This could be a typo error. If not, then the author can explain this.

8)    In their animal study, the authors have taken male mice only. Why have they not included females? Is any reason behind this to exclude female mice (gender biasing)? Authors can explain the gender-specific effect.

9)    Several other probiotic and prebiotic bacteria have been reported for gut health and regulation of metabolic disease including liver disease. How this bacterium is different from other probiotic and prebiotic bacteria? is this bacterium better than other lactobacillus species? 

10) Since this bacterium has already been reported in the regulation of inflammation, dyslipidemia, hypocholesterolemia, insulin resistance, and obesity, as well as modulating gut microbiota. Then what is new in this study?  (References 9 and 15). 

Author Response

Reviewer comments 3

In the current manuscript titled “A Transcriptomic Response to Lactiplantibacillus plantarum-KCC48 against High-Fat Diet-Induced Fatty Liver Diseases in Mice” the authors have determined the transcriptome analysis to establish the correlation between gut microbes (microbiota) and its alteration (dysbiosis) in the progression of high-fat diet-induced Fatty Liver disease. Authors have focused on the role of the specific bacterium Lactiplantibacillus plantarum-KCC48 in the regulation of fat metabolism and regulation of obesity and related fatty liver disease including non-alcoholic fatty liver disease (NAFLD). This is an interesting piece of transcriptome analysis work that will strengthen our understanding in the field of microbiome and related metabolic diseases. However, I have the following comments for this study.

Thank you for your positive comment concerning the submitted research article titled Lactiplantibacillus plantarum-KCC48 against High-Fat Diet-Induced Fatty Liver Diseases in Mice.

  • Although authors have given an explanation that the beneficial bacterium Lactiplantibacillus plantarum-KCC48 is involved in the reduction of fat percentage and adipocytes numbers. Since obesity involves an increase in adipocyte number and size. Thus, did the authors find any explanation regarding the role of Lactiplantibacillus plantarum-KCC48 in the reduction of adipocytes number and size both?

Already we investigated an effect of L. plantarum-KCC48 (L. plantarum A-29) on the expression of adipogenic and lipogenic genes in 3T3-L1 adipocytes and high-fat diet (HFD) -fed mice. We observed that the treatment of 3T3-L1 adipocytes with the cell-free metabolites of L. plantarum inhibited their differentiation and fat depositions via downregulating key adipogenic transcriptional factors (PPAR-γ, C/EBP-α, and C/EBP-β) and their downstream targets (FAS, aP2, ACC, and SREBP-1). Interestingly, supplementation with L. plantarum reduced the fat mass and serum lipid profile concurrently with downregulation of lipogenic gene expression in the adipocytes, resulting in reductions in the bodyweight of HFD-fed obese mice. L. plantarum treatment attenuated the development of obesity in HFD-fed mice via the activation of p38MAPK, p44/42, and AMPK-α by increasing their phosphorylation. Further analysis revealed that A29 modulated gut-associated microbiota composition. Thus, A 29 potential probiotic strain may alleviate obesity development and its associated metabolic disorders.

Ref: Soundharrajan I, Kuppusamy P, Srisesharam S, Lee JC, Sivanesan R, Kim D, Choi KC. Positive metabolic effects of selected probiotic bacteria on diet-induced obesity in mice are associated with improvement of dysbiotic gut microbiota. FASEB J. 2020 Sep; 34(9):12289-12307. doi: 10.1096/fj.202000971R. Epub 2020 Jul 23. PMID: 32701200.

  • How this bacterium involved in the reduction of is inhibits inflammation, dyslipidemia, hypocholesterolemia, insulin resistance, and obesity, as well as modulating gut microbiota? Do authors want to say that these changes in metabolic pathways are related to gut dysbiosis? (https://pubmed.ncbi.nlm.nih.gov/26999104 ).  How does this bacterium affect liver metabolism? Authors can give their supporting explanation if it is available?

Thank you very much for your kind information. The basic mechanisms of probiotics that can be effective in preventing hepatic damages induced by diets have been included in the introduction section with relevant literature.  Please see the following sentences. “Supplementation of probiotics to high fat diet induced obese mice alleviates body weight gain and adiposity by modulating composition of the gut associated microbiota Probiotics and/or prebiotics are effective in lowering serum/lipids levels (Soundharrajan et al., 2020). In animal models, lactobacillus species exhibited potential probiotic and hypocholesterolemia effects (Kim et al., 2017). A number of studies have examined the effects of probiotics on diet-induced NAFLD in animal models. It has been proven that probiotic supplementation, specifically Lactobacillus and Bifidobacterium, can prevent diet-induced fatty liver diseases through downregulation of lipogenesis, reactive oxygen species, proinflammatory markers and mediators, as well as lipopolisaccharide (LPS) and Toll-Like receptor-4 (TLR-4). LPS triggers cytokine cascades and inflammation by interacting with TLR-4. In addition, probiotics increased fatty acid oxidation, antioxidant activity, insulin sensitivity, intestinal mucosal integrity, as well as modulated gut microbiota and bile acid metabolism (Arellano-García et al., 2022; Han et al., 2018)”.

  • Why we are blaming the fat alone for obesity. Current studies ate emphasize the role of sugar, carbohydrates and saturated fatty acids in Fatty liver disease (https://pubmed.ncbi.nlm.nih.gov/34257427/, https://www.sciencedirect.com/science/article/pii/S1551714417306018, https://pubmed.ncbi.nlm.nih.gov/33923255/). Authors should talk about carbohydrates metabolism too in liver disease.

In fact, we strongly agree with the reviewer's opinion that obesity is caused by excessive intake of carbohydrate, excess energy intake, and drinks that are high in sugar. There has been evidence that these nutrients can contribute to the development of obese individuals similarly to excess fat intake; however, these non-fat nutrients will be converted into fat or triglyceride first by the process of denovo synthesis and this will consequently lead to the development of obesity in humans and animals. The development of fatty acid deposition in the liver of mice, a high-fat diet can be used to induce obesity and fatty liver more rapidly in animals than the consumption of other nutrients that are not fat-rich. It is only a reason we have used a high-fat diet for obesity induction. The suggestions made by reviewers shall be considered in our future experiments.  

  • How is this bacterium regulating adipose tissue inflammation? Authors can give their explanation through literature support. In their transcriptome analysis, NLRP12 was increased in the HFD condition as compared to control. What is the status in supplemented group?

There have been a number of studies that have examined the effects of probiotics on diet-induced NAFLD in animal models. Probiotic supplementation, specifically Lactobacillus and Bifidobacterium, has been shown to prevent diet-induced fatty liver disease by inhibiting lipogenesis, reactive oxygen species, proinflammatory markers and mediators, as well as lipopolysaccharide (LPS) and Toll-Like receptor-4 (TLR-4). LPS triggers cytokine cascades and inflammation through its interaction with TLR-4. Furthermore, probiotics increased fatty acid oxidation, antioxidant activity, insulin sensitivity, intestinal mucosal integrity, as well as modulated gut microbiota and bile acid metabolism in humans (Arellano-García et al., 2022; Han et al., 2018). This information’s have been included in the introduction section.

  • As for the expression of NLRP12, NLRP12 was increased in HFD fed animals whereas the expression of NLRP12 was reduced in liver tissues of animals fed with our probiotic at a significant level of only p<05117.However, the supplementary tables included data with significance less than 0.05. Thus, this is only a reason to exclude the NLRP12 expression. Although, as we state in the supplementary tables, expression of NLRP12 in both HFD and HFLPD has been described (NLRP2: 2.160 ± 0.00005 vs 0.850 ± 0.05117). 
  • In figure 1 (Figure 1a and 1b) there is an annotation on the top of the bar c, a b. The author can represent this figure with * and p values. It is not explained in the figure legend.

Thank you. The statistical values have been presented according to the reviewer's suggestions, and changes have been made using red color fonts.

  • The title of the main manuscript and Supplementary figure is different. The authors should correct it.

We would like to thank you for your kind information. Both the manuscript and supplementary files have been revised extensively in response to the reviewer's comments. 

  • In lines 402-403, experimental design section, the authors have written that 30 mice were taken and divided into two groups of 10 mice. I did not understand the groups. This could be a typo error. If not, then the author can explain this.

      The errors have been corrected according to the reviewer's comment in a correct manner.

  •  In their animal study, the authors have taken male mice only. Why have they not included females? Is any reason behind this to exclude female mice (Gender Biasing)? Authors can explain the gender-specific effect.

       An inclusion of only male mice in the study with no female mice would not be biased in any way. There are some natural issues connected with the use of both male and female animals in the same cage that include sexual activity and hormonal changes during the estrous cycle, etc. It is possible that these factors would influence the experimental parameters. However, it will be considered in our future experiment for usage of male and female animals separately.  

  • Several other probiotic and prebiotic bacteria have been reported for gut health and regulation of metabolic disease including liver disease. How this bacterium is different from other probiotic and prebiotic bacteria? Is this bacterium better than other lactobacillus species?

      Yes, we strongly agree with the comment made by the reviewer. According to several studies, Lactobacillus sp can play an important role in the prevention of diet-induced obesity, its metabolic diseases, and disorders. But different Lactobacillus species from different sources exert different effects on diet-induced obesity and fatty liver diseases. 

  •  Since this bacterium has already been reported in the regulation of inflammation, dyslipidemia, hypocholesterolemia, insulin resistance, and obesity, as well as modulating gut microbiota. Then what is new in this study?  (References 9 and 15). 

Yes, we agree with the reviewer's comment. It was reported by Woo Jin Choi et al., in 2020 (Ref 9) that Lactobacillus plantarum LMT1-48 inhibited lipid deposition in 3T3-L1 adipocytes via downregulation of key transcriptional factors and their downstream targets. Therefore, it was suggested that Lactobacillus plantarum LMT1-48 is a potent probiotic that can be used in the treatment of obesity.

Another report by Xianping Li et al., 2020, suggested that the L. plantarum reduced body weight, LPS and inflammatory markers and improved glucose tolerance in animals fed with HFD via regulation of gut associated microbiota.

The present study is also similar to Woo Jin Choi et al., 2020 and Xianping Li et al., 2020, however, in this case we have employed the transcriptome method to examine the global changes in gene expression in liver tissues from HFD and HFDLP fed animals. The source of KCC-48, along with its biological properties such as survival ability in GIT conditions, and antimicrobial activities, might be different compared with previously reported strains.

Arellano-García, L., Portillo, M.P., Martínez, J.A., Milton-Laskibar, I. 2022. Usefulness of Probiotics in the Management of NAFLD: Evidence and Involved Mechanisms of Action from Preclinical and Human Models. International Journal of Molecular Sciences, 23(6), 3167.

Han, R., Ma, J., Li, H. 2018. Mechanistic and therapeutic advances in non-alcoholic fatty liver disease by targeting the gut microbiota. Front Med, 12(6), 645-657.

Kim, S.-J., Park, S.H., Sin, H.-S., Jang, S.-H., Lee, S.-W., Kim, S.-Y., Kwon, B., Yu, K.-Y., Kim, S.Y., Yang, D.K. 2017. Hypocholesterolemic Effects of Probiotic Mixture on Diet-Induced Hypercholesterolemic Rats. Nutrients, 9(3), 293.

Soundharrajan, I., Kuppusamy, P., Srisesharam, S., Lee, J.C., Sivanesan, R., Kim, D., Choi, K.C. 2020. Positive metabolic effects of selected probiotic bacteria on diet-induced obesity in mice are associated with improvement of dysbiotic gut microbiota. The FASEB Journal, 34(9), 12289-12307.

Round 2

Reviewer 1 Report

The manuscript has been improved from the previous version and the authors have clarified all the issues raised. In my opinion, the manuscript can be published in the International Journal of Molecular Science in its current form.

Author Response

Thank you for your positive comment concerning the submitted research article titled Lactiplantibacillus plantarum-KCC48 against High-Fat Diet-Induced Fatty Liver Diseases in Mice.

Reviewer 3 Report

In the revised manuscript, the authors have given a satisfactory explanation and tried to address most of the reviewer comments. However, this study is just adding a new strain of bacterium as a probiotic Lactobacillus sp., and supplementing the previous knowledge which regulates lipid metabolism and obesity-associated liver disease. This study shows the regulation of liver disease by controlling gut microbiota and related metabolism. Thus, the author should emphasize the role of Lactiplantibacillus plantarum-KCC48 in regulating gut microbiota under high-fat diet conditions in preventing fatty liver disease. The author should give a statement at the end of the discussion about what new information was gained from this study that is different from previously established knowledge and limitation of the study.

Author Response

We thank the reviewer for giving valuable comments about our research paper which is very helpful to improve the quality of the presentation of the manuscript. We have gone through the whole manuscript according to reviewer suggestions and modified the same. All changes in the manuscript have been made with red color fonts

In the revised manuscript, the authors have given a satisfactory explanation and tried to address most of the reviewer comments. However, this study is just adding a new strain of bacterium as a probiotic Lactobacillus sp., and supplementing the previous knowledge which regulates lipid metabolism and obesity-associated liver disease. This study shows the regulation of liver disease by controlling gut microbiota and related metabolism. Thus, the author should emphasize the role of Lactiplantibacillus plantarum-KCC48 in regulating gut microbiota under high-fat diet conditions in preventing fatty liver disease. The author should give a statement at the end of the discussion about what new information was gained from this study that is different from previously established knowledge and limitation of the study.

Thank you for your valuable comments and suggestions on the submitted article. We have provided information about our previous work on attenuating diet-induced obesity in mice by P. plantsarum-A29 (KCC-48) and its molecular mechanisms involved in lipid metabolism. Following up on the results of our previous study, we examined global gene expression changes in liver tissues of diet-induced obese mice following L. plantarum-A29 (KCC-48) dietary intervention.

It has been reported that L. plantarum plays a major role in reducing fat mass and its size through regulating genes associated with lipogenesis and fatty acid oxidation via modulating microbiota in GIT. Based on our previous study, we found that L plantarum-A29 (KCC-48) could survive in GIT of diet-induced obese mice, which was confirmed by pyrosequencing. In addition, it reduces adipose tissue mass by downregulating key transcription factors and downstream targets associated with lipid synthesis via pathways including p38MAPK, P44/42, and AMPK-α (Soundharrajan et al., 2020). The number of scientific reports on the effects of probiotic L plantarum on global gene expression in obese animals is still limited. Our knowledge is that this is the first report describing the transcriptome changes in liver tissues of obese animals fed with L.plantarum-KCC-48. The majority of the study's findings are in agreement with the transcriptional changes observed in diet-induced obesity, with the exception of a few studies that contradict ours. Further research will be necessary to re-validate the reported mRNA expressions and their protein expressions using PCR and immunotechniques, respectively. 

Soundharrajan, I., Kuppusamy, P., Srisesharam, S., Lee, J.C., Sivanesan, R., Kim, D., Choi, K.C. 2020. Positive metabolic effects of selected probiotic bacteria on diet-induced obesity in mice are associated with improvement of dysbiotic gut microbiota. The FASEB Journal, 34(9), 12289-12307.